# A Novel, Nontoxic and Scalable Process to Produce Lipidic Vehicles

**DOI:** 10.3390/ma13215035

**Published:** 2020-11-08

**Authors:** Nikolaos Naziris, Natassa Pippa, Costas Demetzos

**Affiliations:** Section of Pharmaceutical Technology, Department of Pharmacy, School of Health Sciences, National and Kapodistrian University of Athens, Panepistimioupolis Zografou, 15771 Athens, Greece; niknaz@pharm.uoa.gr (N.N.); natpippa@pharm.uoa.gr (N.P.)

**Keywords:** modified heating method (MHM), lipidic vehicles, amphiphilic lipids, liquid polyol, physicochemical properties, lyophilization

## Abstract

Lipidic vehicles are novel industrial products, utilized as components for pharmaceutical, cosmeceutical and nutraceutical formulations. The present study concerns a newly invented method to produce lipidic vehicles in the nanoscale that is simple, nontoxic, versatile, time-efficient, low-cost and easy to scale up. The process is a modification of the heating method (MHM) and comprises (i) providing a mixture of an amphiphilic lipid and a charged lipid and/or a fluidity regulator in a liquid medium composed of water and a liquid polyol, (ii) stirring and heating the mixture in two heating steps, wherein the temperature of the second step is higher than the temperature of the first step and (iii) allowing the mixture to cool down to room temperature. The process leads to the self-assembly of nanoparticles of small size and good homogeneity, compared with conventional approaches that require additional size reduction steps. In addition, the incorporation of bioactive molecules, such as drugs, inside the nanoparticles is possible, while lyophilization of the products provides long-term stability. Most importantly, the absence of toxic solvents and the simplicity guarantee the safety and scalability of the process, distinguishing it from most prior art processes to produce lipidic vehicles.

## 1. Introduction

Lipidic-based nanoparticles are clinically and markedly established nanosystems for the delivery of drug molecules, supplements and cosmetic substances, having shown great benefit in the formulation, administration, and delivery of poorly water-soluble bioactive ingredients. Since they are lyotropic liquid crystalline systems, their structure and morphology depend on the concentration of the composing lipids and phospholipids, as well as on their molecular geometry, which is related to their chemical structure [1,2,3]. Their conformation might be lamellar (i.e., a bilayer/membrane), cubic, hexagonal etc., allowing for incorporation of lipophilic, hydrophilic, or even amphiphilic bioactive ingredients. In the case of oral administration, by predissolving these molecules in lipidic vehicles, we overcome the dissolution process, which is usually the rate-limiting step for their absorption in the blood and as a result, for their final effectiveness, especially in the case of lipophilic agents. Lipidic nanoparticles are innovative excipients, presenting variety in their structure and properties in vitro and in vivo, including digestibility and absorption. The nature of their components profoundly affects the final biopharmaceutical and pharmacokinetic profiles of the delivered bioactive ingredients [4].

Among lipidic nanoparticles, liposomes are one of the most promising classes for biomedical applications. They are present in the clinic for years and are still studied, modified, and functionalized, to develop advanced delivery systems for therapeutic molecules. Liposomes are closed pseudospherical structures, consisting of concentric lipid bilayers, which are mainly built by phospholipids and entrap aqueous media. They are inherently thermodynamically unstable in a colloidal dispersion and for this reason, they are usually stabilized through the addition of cholesterol and other types of biomaterials, such as polymers [1,2,3,5,6]. Apart from structural units, these molecules contribute to the final functionality of liposomes, through physicochemical, thermodynamic and biophysical principles. Of great importance for the final liposome properties, stability and functionality is the self-assembly process, which is a thermodynamically driven spontaneous process that is determined by the geometric characteristics and the critical packing parameter of the mixed molecules [7,8].

Liposomes and lipidic vehicles are novel industrial products for pharmaceutical, cosmeceutical and nutraceutical formulations [9,10,11,12,13,14]. The various methods for the production of lipidic vehicles and liposomes include solvent injection, detergent dialysis, freeze-thaw, reverse-phase evaporation, sonication, homogenization and the dehydration-rehydration method, of which one typical technique is the Bangham method (thin-film hydration method). Most of these methods lead to the production of multilamellar vesicles (MLVs), which need to be further processed through size-reduction techniques, so that small unilamellar vesicles (SUVs) are developed, which are utilized for various applications. Another method, microfluidics, is the method of choice for scale-up applications of liposomes and lipid-based systems, with several modifications having been developed. The advantages offered by the microfluidics technology can benefit the production of liposomes and include accurate handling of nanoliter volumes, precise control of the interface position, diffusion-dominated axial mixing and continuous operation for low production volumes [15,16].

According to Patil and Jadhav, 2014, “Conventional techniques for liposome preparation and size reduction’ remain popular as these are simple to implement and do not require sophisticated equipment. However, issues related to scale-up for industrial production and scale-down for point-of-care applications have motivated improvements to conventional processes and have also led to the development of novel routes to liposome formation.” [16]. At the same time, liposomal products are becoming more and more complex, in the attempt of functionalization through surface modification, which leads to integration of more formulation steps, increases the production cost, and renders their evaluation more challenging. For successful scale-up, the prerequisites are few manufacturing steps and the absence of harmful organic solvents. In addition, liposomal development, as part of pharmaceutics, is much dependent on quality assurance and cost. Regarding quality assurance, liposomes, and drug delivery nanosystems in general are affected by production scalability, reproducibility, availability of equipment, expertise, stability of the incorporated bioactive molecule and long-term stability [17]. Moreover, it has been suggested that liposomal drug development can benefit from continuous manufacturing, a processing concept where raw materials constantly flow into a process and product constantly flow out that has been applied to produce biologics [18].

The present investigation is an example of the application of a simple method to produce of lipidic vehicles and liposomes, which is a modified heating method (MHM) (Figure 1) [19,20]. The method comprises stirring a mixture of an amphiphilic lipid and a charged lipid or a fluidity regulator in a liquid medium comprising water and a liquid polyol, heating the mixture in two steps, wherein the temperature of the mixture in the second step is higher than the temperature in the first step and allowing the mixture to cool down to room temperature.

## 2. Materials and Methods

### 2.1. Materials

The phospholipids L-α-phosphatidylcholine, hydrogenated (Soy) (HSPC), L-α-phosphatidylcholine (Egg, Chicken) (EPC) and 1,2-dipalmitoyl-sn-glycero-3-phosphocholine (DPPC), as well as cholesterol (ovine wool, >98%) (CHOL) were purchased from Avanti^®^ Polar Lipids Inc. (Alabaster, AL, USA). Stearylamine (SA) was purchased from Sigma-Aldrich^®^ Co. (St. Louis, MO, USA). All molecules were used without further purification. Glycerine (99.8%), HPLC-grade H_2_O and other reagents were of analytical grade, purchased from Sigma-Aldrich^®^ Co (St. Louis, MO, USA). Curcumin (98%) (CUR) was purchased from Acros Organics (Fair Lawn, NJ, USA). The utilized phospholipids have been established and well-studied in the literature, while they are also components of liposomal products in the market [21,22].

### 2.2. Methods

#### 2.2.1. Lipidic Vehicle Preparation through the MHM

Different types of lipidic vehicles were developed through the present process. HSPC:SA (9:0.25 molar ratio) and EPC:CHOL:SA (9:0.5:0.25 and 9:1.8:0.25 molar ratio) systems were mixed and placed inside a large spherical flask. Purified H_2_O with dissolved glycerine at 20/15/10% *v/v* was subsequently added, to achieve a lipid concentration of 10 mg/mL, followed by brief vortexing of the whole mixture. Afterwards, the mixture was heated at 60 °C and stirred at 700 rpm by utilizing a magnet, for a 1 h period in a silicon oil bath. Then, the formed suspension was heated at 90 °C, while stirring was maintained at the same level, for 1 h. The suspension was cooled down to room temperature at a rate 3 °C/min and a 50 μL sample was extracted for measurement by light scattering. Then, the suspension was heated at 90 °C, while being stirred at 700 rpm for 1 h, and, after cooling at the same rate, another 50 μL sample was extracted for measurement. The final suspension was placed in a glass vial and stored at 4 °C.

HSPC:SA, EPC:SA and DPPC:SA vehicles of molar ratio 9:0.25 and different glycerin concentrations were prepared in samples of 200 mL with the above method, by 1 h heating 60 °C and 2 h heating at 90 °C. Then, they were subjected to 10 cycles of extrusion through polycarbonate filters of pore size 200 nm, by utilizing an EmulsiFlex-C5 (Avestin, Mannheim, Germany), in order to achieve size reduction and homogenization. The extrusion temperature was in each case 1–2 °C above the phase transition temperature (*T_m_*) of the utilized phospholipid. Samples of 50 μL were extracted after cycles 1, 2, 5 and 10 for measurement by light scattering.

#### 2.2.2. Lyophilization and Reconstitution

Lyophilization of the above prepared HSPC:SA 9:0.25 vehicles with 20% or 15% *v/v* glycerine was achieved by utilizing a Cryodos-50 laboratory freeze-drier (Telstar, Barcelona, Spain). Samples were frozen with dry ice and acetone and immediately lyophilized at 10^−2^–10^−1^ mbar. Reconstitution was achieved by adding the same amount of initial purified water volume. Each sample could anneal for 15 min, followed by vortexing, and relaxation period of 15 min.

#### 2.2.3. Incorporation of Curcumin

Curcumin was added to the initial lipidic mixture of HSPC:SA, in molar ratios 9:0.25:0.8 and 9:0.25:1 for HSPC:SA:CUR and the above described lipidic vehicle preparation process followed, including a 2 h heating at 90 °C. Total lipid concentration was between 10 and 50 mg/mL and glycerine concentration varied between 5 and 20% *v/v*. The formulations were placed in glass vials and stored at 4 °C.

#### 2.2.4. Light Scattering Techniques

The developed lipidic vehicles were evaluated in terms of their physicochemical characteristics, by measuring their size (hydrodynamic diameter, *D_h_*), size distribution (polydispersity index, PDI) and zeta potential (z-pot), through dynamic and electrophoretic light scattering (DLS and ELS). To this end, the 50 μL samples that were collected were diluted 60-fold in HPLC-grade H_2_O. The measurements were carried out at a detection angle of 90° and at 25 °C, in a photon correlation spectrometer (PCS) (Zetasizer 3000 HSA, Malvern Panalytical Ltd., Malvern, UK), by measuring the intensity of the scattered light and analyzed by the CONTIN method (Malvern software, V1.6, Malvern Panalytical Ltd., Malvern, UK). Vehicles containing HSPC:SA 9:0.25, EPC:CHOL:SA 9:0.5:0.25 and 9:1.8:0.25, HSPC:SA:CUR 9:0.25:0.8 and 9:0.25:1 were evaluated for their physical/colloidal stability, for around a 30-day period, by taking the formulations out of the 4 °C storage conditions and measuring their size and polydispersity with PCS.

#### 2.2.5. Statistical Analysis

All experiments were performed in triplicate. DLS and ELS results are shown as mean value ± standard deviation (SD) of three independent formulations. Statistical analysis was performed on size and polydispersity results, by utilizing two-tailed paired Student’s *t*-test for sample heating duration at 90° and extruded samples and one-way ANOVA with post hoc Tukey test for sample groups of different glycerine concentration. *p*-values < 0.05 were considered statistically significant. Statistical analyses were performed using SigmaPlot for Windows (Systat Software, Inc., V14.0, San Jose, CA, USA).

## 3. Results

### 3.1. Physicochemical Properties and Colloidal Stability of the Prepared Lipidic Vehicles

Suspensions of lipidic vehicles were developed through the present investigation, composed of HSPC:SA 9:0.25 and EPC:CHOL:SA 9:0.5:0.25 or 9:1.8:0.25 and glycerine 20%, 15% or 10% *v/v* of the total aqueous volume. The results in terms of physicochemical properties are presented in Table 1. Those include the lipidic vehicle size, polydispersity and zeta potential. The hydrodynamic diameter (*D_h_*) is an indication whether the proposed method is efficient in producing systems of small particle size, while the polydispersity index (PDI) regards the homogeneity/heterogeneity of these systems and should preferably be of low value. The distributions by intensity (*D_I_*) of particles with glycerine 20% *v/v*, after 2 h at 90 °C are depicted in Figure 2, while the distributions by intensity (*D_I_*) volume (*D_V_*) and number (*D_N_*) of the same systems built in triplicate are depicted in Appendix A for HSPC:SA 9:0.25 and in Appendix A for EPC:CHOL:SA 9:1.8:0.25. The respective peak analysis, considering peak area, mean and width is given in Appendix A. Finally, the colloidal stability of the systems is illustrated in Figure 3.

From Table 1, we conclude that all present formulations developed through the MHM are roughly in the range of 200 to 400 nm, which in the case of vesicles, corresponds to multilamellar vesicles (MLVs) or large unilamellar vesicles (LUVs). In Appendix A, we observe the repeatable measurements for systems HSPC:SA 9:0.25 and EPC:CHOL:SA 9:1.8:0.25 with glycerin concentration of 20% *v/v*, concerning the intensity, volume and number of particles, which depend on the scattered light. This means that both these systems are stable during the measurement. The distributions are also narrow, indicating systems of low polydispersity. In some cases, dual peaks are observed, especially in the volume diagrams, indicating the existence of two populations of particles of different size. The respective parameter values for each peak are presented in Appendix A.

Concerning the colloidal stability of the systems, their particle size remained stable from day 1 to day 30, with the EPC:CHOL:SA 9:1.8:0.25 system exhibiting an increase on day 15 and then gradually returning to the initial value (Figure 3). The EPC:CHOL:SA 9:0.5:0.25 was unstable.

Systems of different lipidic composition, i.e., HSPC:SA, EPC:SA and DPPC:SA, and glycerine concentration were also developed and subjected to multiple cycles of extrusion, in order to improve their physicochemical properties, i.e., their size and polydispersity. The results are presented in Appendix A.

### 3.2. Physicochemical Properties of Lyophilized Lipidic Vehicles after Reconstitution

The formulations of HSPC:SA 9:0.25 lipidic vehicles with glycerin concentration 20% and 15% *v/v*, developed by 2 h heating at 90 °C, were subjected to lyophilization, reconstitution in the same H_2_O volume and their physicochemical properties were measured by light scattering, in order to compare them with the initial values. The results of the reconstituted samples are provided in Table 2.

By comparing the results in Table 1 and Table 2, the hydrodynamic diameter was preserved in both cases, around 230 nm for glycerine 20% *v/v* and around 290–305 nm for glycerine 15% *v/v*. The polydispersity index was the same for glycerine 20% *v/v* before and after lyophilization, while it was slightly increased for glycerine 15% *v/v*. Finally, the zeta potential was slightly lower for glycerine 20% *v/v*.

### 3.3. Physicochemical Properties and Colloidal Stability of Lipidic Vehicles with Incorporated Curcumin

HSPC:SA 9:0.25 lipidic vehicles with varying concentrations of lipids and glycerine were utilized to incorporate curcumin as a model drug molecule. The results for their physicochemical properties are presented in Table 3, while their 30-day colloidal stability is illustrated in Appendix A. In all these cases, no sedimentation phenomena were observed, indicating incorporation of the molecule inside the lipidic vehicles, since curcumin is an extremely lipophilic molecule and tends to aggregate in aqueous media.

By comparing the physicochemical properties of curcumin-loaded HSPC:SA nanoparticles (Table 3) with neat ones (Table 1) (lipid concentration 10 mg/mL, glycerine concentration 20% *v/v*), we observe that the size was increased by around 50 nm and the polydispersity by 0.2, indicating more heterogeneous particle distribution. Two more formulations with curcumin were developed, with higher molar ratio in curcumin, higher lipid concentration and lower amount of glycerine. The most stable formulations were HSPC:SA:CUR 9:0.25:0.8 and HSPC:SA:CUR 9:0.25:1 of lipid concentration 50 mg/mL and glycerine concentration 10% *v/v*, as we can see in Appendix A.

## 4. Discussion

The role of glycerine concentration in the final properties of the vehicles is evident, where the utilization of higher amounts of the molecule led to HSPC:SA particles of better physicochemical properties, i.e., smaller size, and polydispersity. The liquid polyol drives/enhances the hydration process of the amphiphilic lipid(s), facilitating their self-assembly in smaller vehicles and managing the thermodynamic content of the lipidic vehicles. In addition, glycerine offers several advantages in lipidic vehicle and liposomal formulations. It is a biocompatible, bioacceptable and nontoxic isotonising and dispersant agent for the nanoparticles, enhancing their properties and stability. In addition, it serves as a cryoprotectant during freezing and thawing processes, while its removal from the final product is not necessary [23]. In previous studies, lipidic formulations with EPC and SA, without glycerine were developed by dissolution of the lipids in chloroform, evaporation of the solvent and afterwards, by further processing the final hydrated formulations, in order to achieve the desired physicochemical properties (Table 4). The present approach excludes toxic solvents and size reduction methods that similar studies have utilized in the past, providing a nontoxic, simple, and scalable method for lipidic vehicle preparation. Chloroform and sonication or extrusion through polycarbonate filters are very common elements in liposomal preparation and their avoidance ensures safety and saves time, effort, and costs [24,25,26].

The combination of lipids is also very important for the final physicochemical properties, since HSPC:SA and EPC:CHOL:SA behaved differently, with the CHOL amount inside the system also affecting this behavior. Generally, the process depends on the fluidity/mobility of liquid crystalline materials above their phase transition temperature (*T_m_*), as well as their rigidity below that point. For HSPC and EPC, *T_m_* is ~53 °C and ~−5 °C respectively, which is associated with different self-assembly behavior and membrane stability [27,28]. HSPC is more rigid and stable, while EPC is more fluid and unstable and requires a membrane stabilizer, such as CHOL. As a result, CHOL, as a fluidity regulator, led to physicochemical properties according to its concentration. Finally, the observed positive zeta potential of the prepared lipidic vehicles is attributed to the positive charge of SA (pK_a_~10.65) in the hydration medium, i.e., purified H_2_O with different concentrations of glycerine, where pH is slightly acidic and the amino group of SA is protonated [29].

The different colloidal stability between HSPC and EPC nanosystems is probably associated with the fluid nature of the EPC, compared with HSPC, combined with the multiple thermodynamic transitions it underwent during the measurements (from storage temperature of 4 °C to room temperature and back). As a results, the EPC:CHOL:SA 9:1.8:0.25 nanoparticles rearranged during day 15 and agglomerated due to increased phospholipid fluidity into larger particles, which then disassociated and gradually returned to normal. On the other hand, the EPC:CHOL:SA 9:0.5:0.25 system was unstable, owed to the low concentration of CHOL. The observed colloidal stability of HSPC:SA systems is attributed to the positive zeta potential of these vehicles, which comes from SA and according to the Derjaguin, Landau, Verwey, and Overbeek (DLVO) theory, it leads to electrostatic repulsion between the particles inside the suspension [30,31].

All extruded formulations were above 200 nm before the extrusion process, HSPC:SA being 245.6 nm, EPC:SA 353.5 nm and DPPC:SA 250.2 nm. Interestingly, after only the first pass through the membrane filters, their size was decreased to 135–155 nm, while polydispersity improved as well, especially for EPC (reduction by about 0.350). Then, the following cycles had less effect on the lipidic vehicle properties, mainly homogenizing the HSPC system and reducing the size of the ones with EPC and DPPC. The efficiency of the size reduction method in the pilot scale is very important for future industrial applications [32,33].

The lyophilized systems were concerned to be stable during the freezing, lyophilization and reconstitution processes. In cases where the lipidic vehicles are subjected to lyophilization, the liquid polyol, e.g., glycerine, plays a key role in the physical stability and cryoprotection of lipidic vehicles during lyophilization and reconstitution [34]. Specifically, 20% *v/v* of the molecule led to almost absolute conservation of the physicochemical properties, while 15% *v/v* led to only a slight increase in the particle size and polydispersity, indicating the importance of glycerine concentration levels for this process. An additional advantage of utilizing glycerine as a cryoprotectant is the avoidance of carbohydrates, which are traditionally used for the purpose of lyophilization. Stability of lipidic nanoformulations is imperative for their clinical application and quality, including efficacy and safety. For this reason, these products reach the clinic in their lyophilized form, rendering of fundamental importance the study of the effect of the lyophilization and reconstitution process on their properties, which will in turn affect their biological stability and effect [34].

Curcumin is an extremely lipophilic compound that is characterized by poor bioavailability and rapid metabolism and, as a result, liposomes and lipid nanoparticles are extensively studied as vehicles for the protection and delivery of the molecule [35,36,37,38]. Concerning curcumin incorporation, the herein observed phenomena indicate the interaction of the molecule with the phospholipid bilayer and the possible displacement of SA molecules from the hydrophobic segment of the membrane, leading to alteration of the vehicle physicochemical properties. We also must note the relatively higher concentration of utilized curcumin inside the developed formulations (9:0.25:0.8), compared with past studies. In a previous study, the molecule was incorporated inside EPC liposomes at a 14:1 lipid:drug molar ratio (84% ± 15% incorporation efficiency), leading to a slight decrease in the particle size, accompanied by a slight increase in their polydispersity (Table 5) [39]. In the present study, a comparable case is that of HSPC:SA with glycerin at 20% *v/v*. We observed that, even though the curcumin concentration was higher (around 25% more curcumin), the present MHM works well in producing drug-loaded lipidic vehicles, also considering the important advantages that this method offers, compared with previously established ones.

Concerning the process parameters of the MHM, of crucial importance for the final formulation quality are concentrations, temperature, heating and cooling rate, stirring speed and duration. In Figure 4, we summarize the process parameters that affect the final vesicular properties, i.e., their size and polydispersity. Out of these, we have shown herein that of crucial importance are the lipidic vehicle composition, the concentration of glycerine utilized, the duration of heating the formulation at high temperature and the potential utilization of a size reduction method, such as extrusion. Concerning the method steps, heating the mixture at 60 °C, which is above the *T_m_* of the utilized phospholipids, serves to solubilize the ingredients in the aqueous medium, which are then heated at higher temperature, i.e., 90 °C, in order to form assemblies of small size while in the liquid crystalline phase [23,40]. The duration of heating at high temperature is important and determines the final physicochemical properties of the lipidic vehicles (Table 1). The cooling step is mandatory to be able to measure these properties. The method consistency in producing specific size and polydispersity lipidic vehicles is evident from the measurement standard deviations in Table 1, as well as from the repeatable size distributions in Appendix A.

The present approach offers important advantages in the production of lipidic vehicles and liposomes, compared with other established methods (Table 6) [16]. First, it is nontoxic, by not utilizing chlorinated or other volatile organic solvents. Furthermore, it does not require the use of size reduction methods, such as sonication or centrifugation and leads to particles of adequate size and polydispersity for biomedical applications [41]. In addition, it does not require the use of reduced pressure, it is simple and easy to scale up. The process utilizes the hydration from the aqueous medium molecules and cosolvent molecules, and mechanical shock resulting from the stirring process and heating shock coming from the temperature increase. This leads to the formation of lipidic vehicles and liposomes with desirable characteristics. Through the MHM, it is also possible to incorporate bioactive ingredients inside the lipidic vehicles, such as drugs or active pharmaceutical ingredients (APIs), nutraceuticals, cosmeceuticals etc., by adding them inside the initial liquid mixture, with respect to their thermal stability, hydrophilicity/lipophilicity and other parameters that might affect their incorporation into the lipidic vehicles. It is also possible to subject the final formulations to size reduction methods, such as extrusion and lyophilization methods, to optimize them for various applications.

## 5. Conclusions

The present study describes a simple method, suitable for developing various lipidic vehicles, e.g., liposomes, in the absence of chlorinated and other volatile organic solvents. Prior art processes to produce lipidic nanocarriers have a number of disadvantages, including the utilization of toxic solvents that should be avoided. Furthermore, many prior art processes are complex and/or require considerable amount of resources, time and effort, while the size and homogeneity of the carriers achieved by most of them is not satisfactory and, for this reason, additional steps for size reduction and homogenization are needed. The present investigation provides a process to produce lipidic vehicles which successfully addresses the aforementioned disadvantages. The process can be easily scaled up and utilized in the industry, with or without the use of size reduction techniques, due to simple method parameters, accurate and repeatable results, time efficiency, low cost, absence of chlorinated and volatile organic solvents and safety.

The role of glycerine and its concentration in efficiently producing nanoparticles has been demonstrated, while the composition of the lipidic system is also important. Concerning colloidal stability, HSPC:SA and EPC:CHOL:SA formulations with 20% *v/v* glycerine were stable in due time, however, freeze-drying was carried out, in order to produce stable lyophilized products. The latter were found to maintain their physicochemical properties after reconstitution to the initial volume. The extrusion process applied was also found to be effective in reducing the particle size and homogenizing primarily after the first pass of the vehicles through the polycarbonate filters. Finally, the molecule of curcumin was formulated with lipidic vehicles prepared through the MHM. The proposed method is efficient in producing lipidic nanoparticulate vehicles, which may incorporate bioactive molecules, may be utilized directly, without the use of downsizing techniques and may also be lyophilized to produce stable and consistent products.

## 6. Patents

The present investigation is a complementary document and supports the results of the granted patent EP3533442A1 entitled “Process for the Production of Lipidic Vehicles”.

## Figures and Tables

**Figure 1 materials-13-05035-f001:**
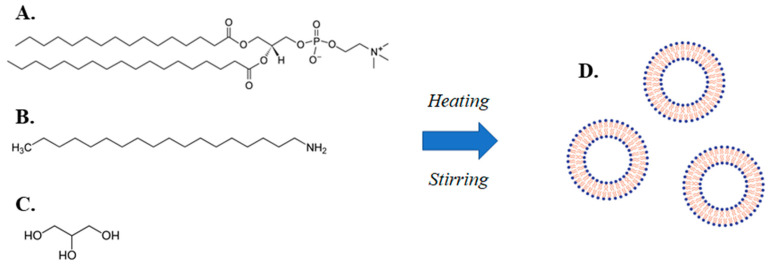
Illustration of the modified heating method (MHM) for lipidic vehicle preparation, including an aqueous mixture of (**A**) a phospholipid (e.g., L-α-phosphatidylcholine, hydrogenated (Soy) (HSPC)), (**B**) a charged lipid or a fluidity regulator (e.g., stearylamine), (**C**) a liquid polyol (e.g., glycerine), which through the application of a two-step heating process accompanied by stirring, leads to the production of (**D**) lipidic vehicles (e.g., liposomes).

**Figure 2 materials-13-05035-f002:**
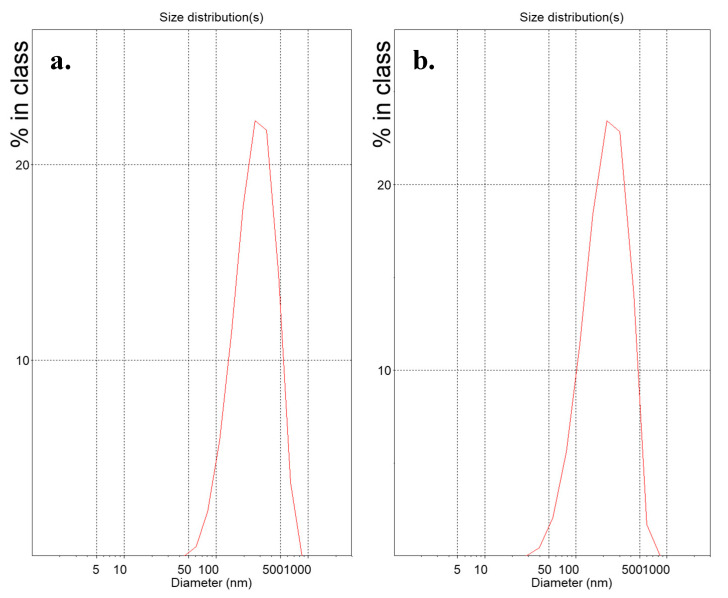
Distribution by intensity (*D_I_*) of (**a**) HSPC:stearylamine 9:0.25 and (**b**) L-α-phosphatidylcholine (Egg, Chicken) (EPC):cholesterol:stearylamine 9:1.8:0.25 lipidic vehicles with glycerine concentration 20% *v/v*. The x-axis represents the nanoparticle size, while the y-axis shows the scattered light intensity%.

**Figure 3 materials-13-05035-f003:**
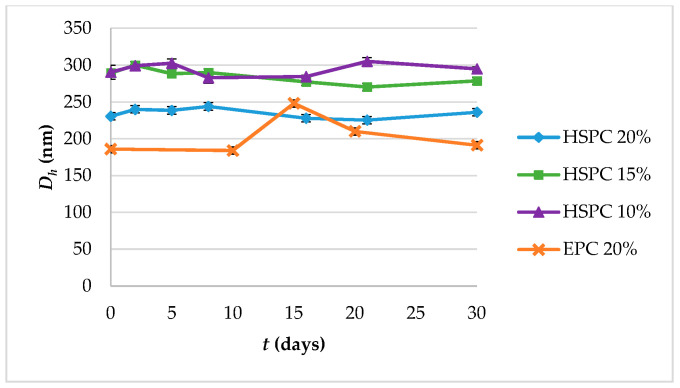
Colloidal stability diagram in terms of particle size (hydrodynamic diameter, *D_h_*) and of systems HSPC:SA 9:0.25 and EPC:CHOL:SA 9:1.8:0.25.

**Figure 4 materials-13-05035-f004:**
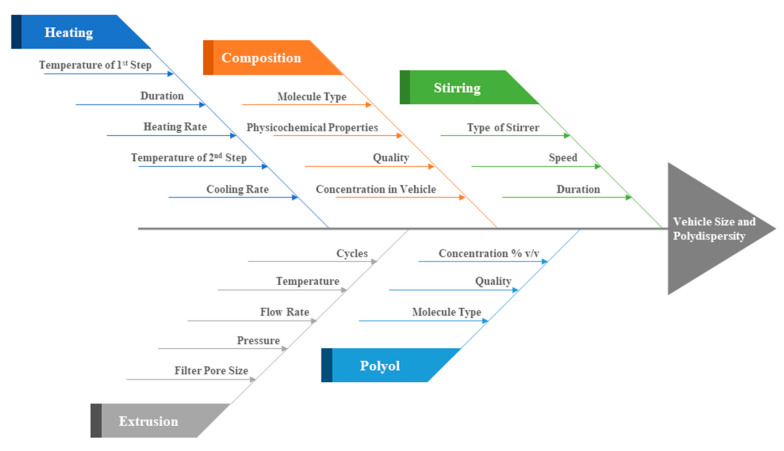
Ishikawa diagram illustrating the main parameters that affect the MHM in terms of lipidic vehicle size and polydispersity.

**Table 1 materials-13-05035-t001:** The physicochemical properties of lipidic vehicles developed through the MHM.

System	Molar Ratio	Glycerine Concentration (% *v/v*)	Hours at 90 °C	*D_h_*^1^ (nm)	SD ^2^	PDI ^3^	SD	z-pot ^4^ (mV)	SD
HSPC:SA	9:0.25	20%	1	275.5	4.1	0.349	0.003	-	-
HSPC:SA	9:0.25	20%	2	230.5	4.6	0.272	0.017	51.5	2.9
HSPC:SA	9:0.25	15%	1	366.8	10.8	0.616	0.078	-	-
HSPC:SA	9:0.25	15%	2	289.2	10.0	0.432	0.016	52.8	2.2
HSPC:SA	9:0.25	10%	1	346.9	9.4	0.567	0.080	-	-
HSPC:SA	9:0.25	10%	2	290.4	9.7	0.484	0.055	54.5	0.6
EPC:CHOL:SA	9:0.5:0.25	20%	2	272.2	13.9	0.602	0.026	-	-
EPC:CHOL:SA	9:1.8:0.25	20%	2	186.0	4.8	0.382	0.007	-	-

^1^ Hydrodynamic diameter; ^2^ standard deviation; ^3^ polydispersity index; ^4^ zeta potential.

**Table 2 materials-13-05035-t002:** The physicochemical properties of reconstituted lipidic vehicles HSPC:SA 9:0.25 with glycerine concentration 20% and 15% *v/v* after lyophilization and reconstitution.

System	Molar Ratio	Glycerine Concentration (% *v/v*)	*D_h_*^1^ (nm)	SD ^2^	PDI ^3^	SD	z-pot ^4^ (mV)	SD
HSPC:SA	9:0.25	20%	228.3	1.6	0.294	0.021	42.3	0.7
HSPC:SA	9:0.25	15%	304.8	1.3	0.559	0.016	49.2	1.9

^1^ Hydrodynamic diameter; ^2^ standard deviation; ^3^ polydispersity index; ^4^ zeta potential.

**Table 3 materials-13-05035-t003:** The physicochemical properties of lipidic vehicles with incorporated curcumin developed through the present investigation.

System	Molar Ratio	Lipid Concentration (mg/mL)	Glycerine Concentration(% *v/v*)	*D_h_*^1^ (nm)	SD ^2^	PDI ^3^	SD
HSPC:SA:CUR	9:0.25:0.8	10	20%	277.9	5.3	0.468	0.056
HSPC:SA:CUR	9:0.25:1	33	5%	459.3	11.5	0.842	0.032
HSPC:SA:CUR	9:0.25:1	50	10%	539.4	5.8	0.706	0.126

^1^ Hydrodynamic diameter; ^2^ standard deviation; ^3^ polydispersity index.

**Table 4 materials-13-05035-t004:** The physicochemical properties of lipidic vehicles with incorporated curcumin developed through the MHM and other methods.

System	Preparation Method	Molar Ratio	*D_h_*^1^ (nm)	SD ^2^	PDI ^3^	SD	Source
EPC:CH:SA	MHM	9:0.5:0.25	272.2	13.9	0.602	0.026	Present
EPC:CH:SA	MHM	9:1.8:0.25	186.0	4.8	0.382	0.007	
EPC:CH:SA	TLE ^5^	6.6:10.3:3.71	408.6	279.1	0.56	0.01	24
EPC:CH:SA	TLE	6.6:10.3:7.42	311.4	154.6	0.25	0.09	
EPC:CH:SA	TLE	6.6:10.3:11.13	348.7	100.3	0.28	0.24	
EPC:CH:SA	REV	6.6:10.3:3.71	313.5	97.8	0.30	0.15	
EPC:CH:SA	REV	6.6:10.3:7.42	376.5	160.7	0.18	0.17	
EPC:CH:SA	REV	6.6:10.3:11.13	603.7	180.5	0.34	0.31	
EPC:CH:SA	FAT ^6^	6.6:10.3:3.71	360.0	220.3	0.48	0.12	
EPC:CH:SA	FAT	6.6:10.3:7.42	351.1	121.8	0.31	0.15	
EPC:CH:SA	FAT	6.6:10.3:11.13	317.1	232.8	0.26	0.14	
EPC:SA	REV ^4^	10:0.1	91.3	0.7	-	-	25
EPC:SA	TFH ^7^	95:5	131	8	0.082	0.08	26
EPC:SA	TFH	90:10	127	2	0.098	0.01	
EPC:SA	TFH	85:15	123	3	0.112	0.01	

^1^ Hydrodynamic diameter; ^2^ standard deviation; ^3^ polydispersity index; ^4^ reverse phase evaporation; ^5^ thin-layer evaporation; ^6^ freezing-thawing; ^7^ thin-film hydration.

**Table 5 materials-13-05035-t005:** The physicochemical properties of lipidic vehicles with incorporated curcumin developed through the MHM and the thin-film hydration method (TFH) method.

System	Preparation Method	Molar Ratio	Glycerine Concentration (% *v/v*)	*D_h_*^1^ (nm)	SD ^2^	PDI ^3^	Source
HSPC:SA	MHM	9:0.25	20%	230.5	4.6	0.272	Present
HSPC:SA:CUR	MHM	9:0.25:0.8	20%	277.9	5.3	0.468	
EPC	TFH ^4^	-	-	123.5	9.9	0.100	39
EPC:CUR	TFH	14:1	-	108.0	8.9	0.146	

^1^ Hydrodynamic diameter; ^2^ standard deviation; ^3^ polydispersity index; ^4^ thin-film hydration.

**Table 6 materials-13-05035-t006:** Advantages and disadvantages of various methods that produce lipidic vehicles and liposomes.

Production Method	Absence of Toxic Organic Solvents	Simple	Small and Monodisperse Vehicle Size	Does not Require Size-Reduction	Easy Scale-Up
Thin-film Hydration	-	✓	-	-	-
Reverse-phase Evaporation	-	✓	-	-	-
Solvent Injection	-	✓	-	-	✓
Detergent Depletion (Dialysis)	-	-	-	-	-
Supercritical Fluid	✓	-	-	-	✓
Modified Heating Method	✓	✓	✓	✓	✓

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
