# Peer review of "A Novel, Nontoxic and Scalable Process to Produce Lipidic Vehicles"

_materials, 2020, doi:10.3390/ma13215035_

Round 1

Reviewer 1 Report

This is a research article and not a patent, then special attention should be dedicated to the fundamental aspects. All presented results should be deeply discussed and well illustrated. 

PDI should be removed and replaced by standard deviation. 

Figure 2-B is not necessary. No scientific importance. 

The manuscript is free from any size distribution illustration. It will be good to have one illustrations.

What is the origin of the positive zeta potential. This should be discussed in the manuscript. In addition at which pH and salinity.

Zeta potential as a function pH will be of paramount importance. Why this wasn't investigated?

What is the key parameter affecting the particles size?

Most discussion should be dedicated to colloidal characteristics before and after lyophilisation and reconstitution. This should be deeply discussed.

Conclusion is too light and should be more consistant.

Author Response

The answers are included in the statement of corrections.

Reviewer 2 Report

Line 108 - The author’s state that HSPC and EGGPC were mixed with stearylamine, the authors should clarify if both lipids were mixed together or if one sample contained HSPC/SA and another contained EGGPC/SA.

Line 145 – The author’s state DLS and ELS experiments were performed in triplicate (N = 3). Is this three measurements of a single preparation of lipid particles, or measurements of three preparations of lipid particles? Were the generation of lipid particle done more than once, and if so how consistent is the methodology?

Figure 2. The authors should state what weighting (i.e. intensity, volume or number) they are reporting.

The authors should state the efficiency of curcumin encapsulation

The authors should provide additional characterization information on the lipid particle they have produced. Are these solid lipid nanoparticles, multilamellar vesicles or unilamellar vesicles generated with MHM?

The authors state that quality of a lipid particle is dependent and the size and polydispersity (line 212-213) yet in table S2 the size and PDI of HSPC:SA 20% Glycerine  particles prior to extrusion is less than a similar lipid particle formulated with their MHM methodology for 1h and similar to those processed after 2 h. For this reviewer, this brings into question the utility of their method. The authors need to discuss some additional benefit that the MHM method brings, i.e. producing unilamellar vesicles etc…

The authors should provide the starting diameters of the particles prior to MHM

Do the authors have any data on the stability of the lipids when heating at 90C for 2 hours? How does this high temperature affect loading of drugs etc.?

Author Response

The answers are included in the statement of corrections

Reviewer 3

Comment

Line 108 - The author’s state that HSPC and EGGPC were mixed with stearylamine, the authors should clarify if both lipids were mixed together or if one sample contained HSPC/SA and another contained EGGPC/SA.

Answer

The particular segment has been modified to be clearer and more specific (Lines 107-109).

Comment

Line 145 – The author’s state DLS and ELS experiments were performed in triplicate (N = 3). Is these three measurements of a single preparation of lipid particles, or measurements of three preparations of lipid particles? Were the generation of lipid particle done more than once, and if so, how consistent is the methodology?

Answer

The experiments were all performed in triplicate.  A statistical analysis part has been added (Lines 150-156).  DLS and ELS results are shown as mean value ± standard deviation (SD) of three independent formulations. The consistency of the method in producing specific particle size and polydispersity is reflected on the standard deviation of the measured size and polydispersity (Table 1) and on the Figure S1 distributions (Lines 327-329). The method has been also granted the European Patent Number EP3533442A1 and will be utilized due to its consistency for the industrial production of lipidic vehicles.

Comment

Figure 2. The authors should state what weighting (i.e. intensity, volume or number) they are reporting.

Answer

The weighting of the method is by intensity and is now stated in the methods section and regards all the measurements (Lines 144-145).

Comment

The authors should state the efficiency of curcumin encapsulation.

Answer

Curcumin is an extremely lipophilic bioactive compound, which we utilized as a model molecule. In the experimental procedure, we did not observe any sediments of the curcumin in the vial (Lines 219-212). In addition, the encapsulation efficiency for similar formulas has been established in our lab’s previous publications (around 85%) and was not the aim of this study. The aim of this study was to present an improved method that produces lipidic vehicles and may be utilized for the encapsulation/incorporation and delivery of bioactive molecules.

Comment

The authors should provide additional characterization information on the lipid particle they have produced. Are these solid lipid nanoparticles, multilamellar vesicles or unilamellar vesicles generated with MHM?

Answer

The method that is described in this manuscript is used to prepare lipid vehicles. According to the physicochemical characteristics of the prepared systems (Tables 1, 2 and 3), the size of the vehicles ranges between 100 and 600nm. Generally, this size is referred to multilamellar vesicles (MLVs) or large unilamellar vesicles (LUVs). In this study, we gave our attention to physicochemical characteristics of the prepared lipid vesicles and not to their morphological characteristics. Please check Lines 183-185.

Comment

The authors state that quality of a lipid particle is dependent and the size and polydispersity (line 212-213) yet in table S2 the size and PDI of HSPC:SA 20% Glycerine particles prior to extrusion is less than a similar lipid particle formulated with their MHM methodology for 1h and similar to those processed after 2 h. For this reviewer, this brings into question τhe utility of their method. The authors need to discuss some additional benefit that the MHM method brings, i.e. producing unilamellar vesicles etc.

Answer

The systems that were subjected to extrusion were prepared by a 2-hour heating process at 90°C. For this reason, the HSPC:SA 20% system intended for extrusion is similar to the one that was developed initially. Please refer to Line 120-121.  Regarding the additional benefit about the morphology of the prepared lipid systems, please see the answer of the previous comment.

Comment

The authors should provide the starting diameters of the particles prior to MHM.

Answer

According to our method, there are not any formed particles prior to the MHM.

Comment

Do the authors have any data on the stability of the lipids when heating at 90°C for 2 hours? How does this high temperature affect loading of drugs etc.?

Answer

According to the reviewer’s comment, we enriched our references with the manuscript entitled: Stability of lipid domains FEBS Letters 584 (2010) 1653–16. At 90°C all the lipids are in liquid crystalline phase (Lines 322-325). The temperature probably affects the loading of the active pharmaceutical ingredient, but this is a purpose of a future study. In this manuscript, we used curcumin as a model molecule/model drug to prove that our method can be used for the incorporation of drugs into lipid vehicles. For each drug, several parameters should be considered i.e. lipophilicity, logP, pKa, thermal stability etc. in order to use this method as preparation protocol of lipid vehicles. The same parameters are taken into account for other methods as well, e.g. thin-film hydration process where the lipid/drug mixture is heated above the Tm of the lipid (Lines 344-345).

Reviewer 3 Report

1) The authors propose a newly invented method to produce liposomes. They have written that the process is a modification of the heating method and comprises ... heating the mixture in two heating steps. At the first step, the mixture was heated to 60 ºC and stirred for 1 h. Then the formed suspension was heated to 90 ºC and also stirred for 1 h. After that, the suspension was cooled to room temperature at a rate of 3 ºC/min. Then, the suspension was again heated to 90 ºC and stirred for 1 h, and again cooled at the same rate.

The necessity of all these steps is not explained in the article. What size liposomes will be formed if the system is heated up to 90 ºC at the first step? How much does the liposome size change after the second heating and cooling?

The text of the article should not repeat the text of claim in the patent. If the method as new, then its description should be more detailed with an explanation of the physicochemical processes occurring in the system at each stage.

2) The authors regard that this multistage process leads to the self-assembly of nanoparticles of small size. What is mean by nanoparticles that self-assembled?

3) Why did liposome sizes in the EggPC system increase on day 15 and then return to the initial value?

Author Response

The answers are included in the statement of corrections

Reviewer 2

The authors propose a newly invented method to produce liposomes. They have written that the process is a modification of the heating method and comprises ... heating the mixture in two heating steps. At the first step, the mixture was heated to 60°C and stirred for 1h. Then the formed suspension was heated to 90°C and also stirred for 1h. After that, the suspension was cooled to room temperature at a rate of 3°C/min. Then, the suspension was again heated to 90°C and stirred for 1h, and again cooled at the same rate.

Comment

The necessity of all these steps is not explained in the article. What size liposomes will be formed if the system is heated up to 90°C at the first step? How much does the liposome size change after the second heating and cooling?

Answer

The present method is a variation of the heating method. The utilization and usefulness of these steps are discussed in Lines 322-327, with reference to that method. After the second heating-cooling cycle, the physicochemical properties are improved in terms of size, with 50-80nm reduction, and polydispersity.

Comment

The text of the article should not repeat the text of claim in the patent. If the method as new, then its description should be more detailed with an explanation of the physicochemical processes occurring in the system at each stage.

Answer

We did major revisions through the manuscript. According to the reviewers’ comments, we added detailed explanation of the physicochemical processes.

Comment

The authors regard that this multistage process leads to the self-assembly of nanoparticles of small size. What is mean by nanoparticles that self-assembled?

Answer

The self-assembly process is mentioned as the spontaneous thermodynamic mechanism that takes place when adding the amphiphilic molecules to aqueous medium and subjecting them to mechanical and thermal processes, i.e. stirring and heating. It is explained in this regard in Lines 49-52.

Comment

Why did liposome sizes in the EggPC system increase on day 15 and then return to the initial value?

Answer

This is an effect that relates to the fluidity of the EPC phospholipids compared with HSPC and probably, to the consecutive freezing and de-freezing of the system in order to measure it each day. Please refer to Lines 256-263 and 267-273.

Round 2

Reviewer 1 Report

The authors have considered are suggested comments and modifications.

The manuscript is now acceptable as such.

Reviewer 3 Report

All comments are corrected.